# Stakeholder Management in Government-Led Urban Regeneration: A Case Study of the Eastern Suburbs in Chengdu, China

Jue Wang [1,2,*] , Yi Yang [3], Huan Huang [1,2,4] and Fan Wang [1]

1   Business School, Chengdu University of Technology, Chengdu 610059, China; huan@cdut.edu.cn (H.H.); wangfan@stu.cdut.edu.cn (F.W.)
2   Digital Hu's Line Research Institute, Chengdu University of Technology, Chengdu 610059, China
3   College of Management Science, Chengdu University of Technology, Chengdu 610059, China; yangyi1@stu.cdut.edu.cn
4   State Key Laboratory of Geohazard Prevention and Geoenvironment Protection, Chengdu University of Technology, Chengdu 610059, China
*   Correspondence: wangjue19@cdut.edu.cn

**Abstract:** There are debates on choices between the bottom-up and top-down urban regeneration approaches, and the former is often considered to be better quality since it includes mass stakeholders. This research aims to extend the understanding of the government-led top-down urban regeneration approach with a case study of the Eastern Suburbs in Chengdu, China. A qualitative interview-based approach was adopted. The results show that the top-down approach is efficient and brings high-quality results for large-scale post-industrial transformation with inclusive stakeholder management. There are several possible ways to involve stakeholders: actively or passively, participating in the whole process or some stages, and engaging fully or partially, according to the demand on site.

**Keywords:** government-led urban regeneration; post-industrial transformation; stakeholder management; top-down approach



## 1. Introduction

Urban regeneration is often considered a tool for renewing post-industrial regions, transforming economies, and promoting sustainable social and environmental agendas [1–5]. The experience of post-industrial towns in Europe and North America has shown that urban regeneration could neutralize the detrimental effects during deindustrialization with the reorientation to reusing the abandoned areas and buildings through improvement and integration [6–8]. For instance, the Tate Modern Gallery in England was transformed from a power plant [9]. Such reuse makes the most of existing land and resources and contributes to sustainability through environmental and ecological remediation, economic reintegration, social revitalization, and cultural heritage protection [10,11].

Since the late 1990s, China has experienced rapid economic restructuring and continuous urban expansion. As a result, the industrial zones and factories in the city were eliminated or replaced [12], and have opportunities to rejuvenate into service-oriented facilities such as shopping malls, theaters, museums, office spaces, city parks, and commercial complexes [4,11]. Urban regeneration has attracted government attention to upgrade abandoned industrial sites and achieve coordinated and sustainable development [13]. Related to the creative clusters boom, the systematic adaptive reuse of derelict industrial sites has become an essential task in urban regeneration in China [14–16]. For instance, in Beijing, the famous 798 Park was transformed from an industrial base to a creative park attracting cultural industry [17]; in Shanghai, the Red Town was transformed from industrial properties to spectacular buildings and served as office spaces [18].

Although abundant western cases are showing the importance and advantages of the bottom-up approach, it is noted that the concept of western urban regeneration might not be mapped directly onto the eastern contexts due to cultural and political differences [3]. We found that the advantages and disadvantages of top-down and bottom-up are widely discussed, but the best use cases for both methods are rarely mentioned. Looking at local cases under eastern contexts could replenish the current understanding of urban regeneration approaches and enrich circumstances and knowledge of existing urban regeneration models.

In recent decades, China's rapidly growing economic transition has accelerated government-led urban regeneration. Government intervention works with market forces to realize capital accumulation through land reuse under the combined action of socialist history and current global trends [19]. Government departments often act as the most important stakeholders, establish rules and systems for urban renewal and directly affect the participation of other stakeholders [20]. In such practices, public participation is often a critical issue grasping the attention of society and academia. Some researchers criticized that urban regeneration led by the Chinese government usually contains minimal stakeholder involvement [21], and some found that stakeholders' interests were sufficiently considered leading to successful results of the regeneration. For instance, Zhuang et al. conducted stakeholder analysis in a case study of Chongqing city regeneration, considering stakeholders' knowledge, power, interests, as well as their relationships and network structures [22]. In a regeneration project in Shanghai, a three-population evolutionary game framework was used to analyze the interests of the government, developers, and residents, resulting in a flexible subsidy scheme to stimulate cooperation between developers and residents and reduce the excessive financial burden on the government [23]. In an urban village renovation project in Shenzhen, the cause and process of community stakeholders losing discourse rights were concerned, thus promoting public participation to ensure the interests of all stakeholders [24]. These practices emphasize the collective cooperation among government, private sectors, and communities to address multifaceted sustainability issues.

Current studies have provided a clear picture of who the stakeholders are and how they relate to each other in government-led urban regeneration projects. However, the existing literature often focuses on the consequences, with little attention paid to the process [25], particularly how the stakeholders bargain and interact in different environments [14]. Therefore, this research takes the urban regeneration of the post-industrial Eastern Suburbs in Chengdu, China, as a case study to review its top-down regeneration process from 2000 to 2020, with particular attention paid to stakeholder management. We mainly consider three questions: How did stakeholders participate and collaborate in the reconstruction process? Is top-down government leadership effective? How to integrate stakeholders' interest in the government-led process?

Following this section, the Section 2 reviews the literature, the Section 3 introduces the case study and research methods, the Section 4 reviews the historical process and effects of Eastern Suburbs regeneration, and the Section 5 analyzes the stakeholder management in the regeneration process. Sections 6 and 7 are discussion and conclusion, respectively.

## 2. Literature Review

Urban regeneration has inspired many literature and practices; however, the challenge is to agree on operating in an area subjected to changes. In terms of the management mode, there are two main approaches—"bottom-up" and "top-down" [26]. The bottom-up approach is seen as a quality approach that enhances public engagement, increases social cohesion, enlivens local identity, and bolsters the legitimacy of urban regeneration projects [27–29]. It focuses more on local communities to develop plans and then promote the implementation of recommendations [30], often with more emphasis on community empowerment than the top-down approach. Without strong government involvement, it means more public participation and effectiveness [31–33]. Since urban regeneration seeks

comprehensive strategies considering simultaneous adaption of multi-perspectives [21,34], participation and empowerment are often considered positively correlated with quality decision making, identification and solidarity of the community, and social well-being [35–37]. However, it may exclude potential stakeholders due to elite capture, especially in areas with severe information asymmetry, imperfect regulatory mechanisms, and low levels of democracy [38,39]. In addition, the full engagement required in the bottom-up approach is hardly achieved in practice [26].

On the contrary, the top-down regeneration mode is a hierarchical mode characterized by strong government force and weak public participation [40]. Communities are often seen as recipients of project outcomes rather than partners in project implementation and design [41]. The top-down approaches tend to be results-oriented and enhance project processes by leveraging local knowledge and resources, often involving partnerships with community leaders and higher-level institutions, followed by interventions at the local level [42]. The government plays a leading role from a macro perspective and allocates resources effectively with its power [20], which could largely reduce the uncertainties faced by business participants in land reuse and redevelopment [19]. Compared with western systems, top-down regeneration in China is a powerful and legitimate means of releasing non-commoditized property [43]. Urban regeneration emphasizes the multi-party cooperation among government, the private sector, and communities to improve human settlements and preserve the community's resources from multiple perspectives [44]. However, the top-down approach may cause ignorance of community dynamics and absence of transparency, and have been criticized for failing to consider local circumstances and generating irrelevant and/or inappropriate interventions [14,28,45–47]. The reasons were multiple, including institutional arrangement, complex social networks, and sometimes stakeholders' negative perceptions towards participation [22].

Needless to say, full-fledged public participation is ideal. Broad public participation can provide actionable urban solutions based on local knowledge and enable policymakers to make better decisions [48]. The value of public participation in strengthening democracy and maintaining legitimacy has long been recognized by scholars and practitioners [49,50]. However, community engagement has encountered many obstacles in urban management in China, including centralized governance, marketization, and lack of expertise among residents [21,51]. Many scholars have pointed out the gap in public participation between developing and developed countries and realized that there is no universal public participation mechanism [52]. Some scholars have suggested that when we look at developing countries, participatory planning theories formed and developed in Western countries often lack consideration of the potential impact of political, social, and economic contexts [52,53]. The "institutional context" of public participation should be fully considered. Participatory planning approaches can only be successfully applied in developing countries if the political and social context is considered [54].

Compared with developed countries with a long history of democracy, the practice of public participation in China lags due to multiple reasons, particularly its political characteristics [55]. Sometimes, despite the government's efforts to increase public participation, an equal rights approach for all stakeholders may not enhance the decision-making power of any group other than the government [56]. In China, inclusive and participatory governance of urban heritage is still limited, and effective ways to involve the public in decision-making remain to be fully explored [51]. In community planning, the regulation of the state and the government is dominant, and the community lacks a general social foundation and corresponding institutional support [57]. Community engagement in China is still in its infancy, showing its contextual characteristics and needs to be made more inclusive and to a higher degree by learning global approaches and localizing them to suit Chinese conditions [51].

## 3. Methodology

Case study often contains in-depth information and provide an insightful understanding of the social phenomenon. In this article, we take Eastern Suburbs in Chengdu, China, as a case study to unfold the government-led post-industrial urban regeneration process under the Asian context, particularly paying attention to stakeholder management. The renewal of the Eastern Suburbs was initiated by local government through a top-down approach in 2000. The main body of the industrial area opened as a cultural park in 2012, and some parts continued to be renewed and improved until today.

Under the umbrella of qualitative research, we have adopted field study, participant observation, and semi-structured interviews for data collection during January to April 2020. To evaluate the situation and effects of the reconstruction of the Eastern Suburbs, a field study and participant observation were applied to generate the direct view and firsthand information. Such methods give the researchers intimate familiarity with the research area and people involved in the environment. Considering that the urban regeneration process is closely related to the subjective view of various stakeholder groups, semi-structured interviews with 42 stakeholders in six groups were conducted (Table 1), providing in-depth information to understand the topic. The semi-structured interview questions were developed based on the previous informal talks and discussions with local officials, tourists, residents, etc. Quantitative measures such as ranking and scoring were integrated into the interviews. Historical documents and images were collected during the interviews when possible. The interviews were audiotaped with respondents' permission and then transcribed into text. We confirmed that the interviewee was free to express.

**Table 1.** List of interviewees.

| Stakeholder Groups | Nr. of Interviewees | Contents of Interview |
| --- | --- | --- |
| Government staffs | 2 | The historical process of Eastern Suburbs regeneration <br> Key stakeholders and their roles and functions <br> Conflicts and solutions among stakeholders |
| Park managers | 2 | Initiation and construction process of the Eastern Suburbs Memory Park <br> The operation status of the Eastern Suburbs Memory Park |
| Shop staff | 5 | The reason setting up the store in the Eastern Suburbs Memory Park <br> Statement of operations: guest flow, sales volume, income |
| Factory former employees | 6 | Resettlement and employment after the regeneration |
| Residents | 12 | Sense of the place in the renewed Eastern Suburbs <br> Satisfaction living around the Eastern Suburbs |
| Tourists | 15 | Experience and impression on the renewed Eastern Suburbs <br> Evaluation of the facilities and services provided in the Eastern Suburb Memory Park |

Qualitative content analysis was applied to analyze the transcripts with the software MaxQDA2020. We first carried out open coding, widely discovered conceptual categories from texts, named them to determine the attributes and dimensions of the categories. Then, we verified the consistency of the attributes and the encoding categories and refined the

encoding. Finally, we extracted the encoded text in the encoding that fits our topic and presented the results with diversified means, e.g., word cloud. Due to the limitations of case studies, we need to be cautious and comprehensively consider the specific conditions of different regions when generalizing the conclusions.

## 4. The Case of Eastern Suburbs Urban Regeneration

This section may be divided by subheadings. It should provide a concise and precise description of the experimental results, their interpretation, as well as the experimental conclusions that can be drawn.

### 4.1. The Background of Eastern Suburbs Urban Regeneration

The Eastern Suburbs were constructed in the 1950s as an industrial base occupied 40 km$^2$ land with more than 160 large-scale electronics, machinery, and metallurgy enterprises. Since the 1990s, China's economic development has shifted from chronic shortage to relatively surplus industrial products with a low production capacity utilization rate. The production capacity of most products exceeds market demand, leading to excessive competition and waste of resources in the processing industry. In 1990, the gross output value of the Eastern Suburbs industrial zone accounted for more than 75% of the city's total state-owned industrial output value. During the 1990s, these enterprises faced declining profits, increasing debt, bankruptcy, and mass lay-offs with Chinese economic transformation. Severe urban pollution gradually emerges and aggravates due to industrial emissions and sewage discharge. With the rapid expansion of urban space, the Eastern Suburbs have become an essential part of urban development, and the factories seem to be out of place. These challenges called for the regeneration of the old industrial zone. Since 2002, China has entered a stage of rapid urbanization. At this point, cultural and creative industries have become a new economic growth point. With the marketization of the economy, the creative industry flourishes under the needs of upgrading, technological progress, globalization, and economic restructuring.

### 4.2. The Process of Eastern Suburbs Urban Regeneration

The discussion on the Eastern Suburbs urban regeneration sprouted in 2000. In January 2002, the city government issued "Interim Measures for the Relocation and Reconstruction of Industrial Enterprises in Eastern Suburbs," initiating large-scale transfer and relocation of industrial plants from the Eastern Suburbs to farther outskirts, aiming at structural economic reform and environmental improvement. In 2006, the "Eastern Transfer" was completed. According to the "Protection Plan for Chengdu Excellent Modern Buildings" promulgated by the city government, the abandoned factory buildings were preserved as industrial heritage, constituting the industrial civilization relics of the planned economy and the early stage of the market economy. According to the "Chengdu Cultural and Creative Industry Development Plan 2009–2012", the government decided to reconstruct Eastern Suburbs into a cultural and creative industry base. The state-owned Hongguang Electronic Factory site was selected and agreed to be rebuilt as an industrial heritage park—the Eastern Suburb Memory Park (Figure 1). In July 2009, Chengdu Media Group was established to be fully responsible for the planning and construction of the park, with an investment of five billion yuan. It opened in 2012, retaining some old factory buildings and landmarks to maintain the industrial culture and history. It has soon become a multifunctional tourist attraction that combines industrial heritage and creative cultural elements—music, art, drama, photography, and other artistic forms. Some parts of the regeneration continue today, and many stakeholders have been involved, such as residents, tourists, service providers, and small businesses. Some features of the site continue to update to this day. The regeneration has turned an industrial zone into a multifunctional space, including residential, commercial, leisure, and cultural functions.

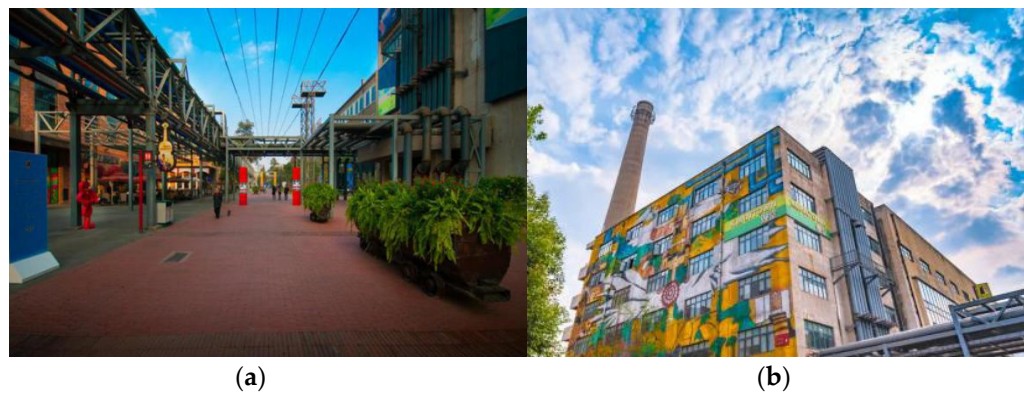

|  (a)  |  (b)  |

**Figure 1.** The Eastern Suburb Memory Park: (**a**) the central avenue; (**b**) the old factory building serving as an office building (photo taken by the authors).

### 4.3. The Effects of Eastern Suburbs Urban Regeneration

After comprehensive and systematic renewal, the overall appearance of Eastern Suburbs has undergone tremendous changes, bringing considerable environmental, economic, cultural, and social benefits.

#### 4.3.1. The Environmental Effects

To tackle environmental problems caused by yearly industrial emissions, an innovative solution, namely "treatment–restoration–greening," was introduced for ecological renewal. First, it dealt with abandoned industrial land and polluted waterfront areas, established harmful gas and dust recovery devices, and controlled air pollution. Then, ecological restoration has become the primary means to change the appearance of the old industrial area, which is possible to avoid the removal of abandoned industrial facilities, conserve energy, protect the environment, and save resources. In addition, systematic landscape planning and garden design allow greening the formerly desolate industrial area by preserving the original plants, adding new vegetation in the reconstruction area, and rebuilding its own ecological system. The green area covers 46804 m2, accounting for 32.2% of the total area.

#### 4.3.2. The Economic Effects

The economic benefits include the profit from the enterprises' relocation and reconstruction of tourist attractions and creative cultural industries. After the Eastern Transfer, the industrial concentration degree and its agglomeration effect of the relocation has gradually emerged. The industrial layout has become more reasonable, effectively promoting regional sustainable development. Economic participants often bring profit and investment in commerce, retail, and other services [58,59]. By December 2019, the Eastern Suburbs has cultivated seven significant forms—performing arts, audio-visual performance, bars and recreational settings, theme retail, catering, theme hotels, business office, and training formats, with a total of 185 companies. There are five million visitors to the park every year and exceeds 30 million cumulatively. A large number of the literature shows that the regionalization of the cultural and creative industry is driven mainly by the economic interests of local governments, which obtain a large number of profits from land development and use it to control cultural production [60]. The economic value brought by the cultural and creative industries to the whole city is also very considerable, increasing from 6.581 billion yuan to 145.98 billion yuan from 2004 to 2019.

#### 4.3.3. The Cultural Effects

The effect of urban regeneration should not only be expressed in terms of the physical environment and economic benefits; the intangible impacts in cultural and social aspects are also essential [4,61,62]. As one of the important industrial areas in China, Eastern

Suburbs' historical and cultural value is inestimable. The renovation combines protection with creativity to continue the historical context of the area. People can feel the renewal of the site and the charm of old industrial buildings, and the ubiquitous creative sketches of rusty pipes, waste boilers, and steel frames. The Eastern Suburbs is the first real-scale transformation of the post-industrial area in Chengdu, highlighting the historical and cultural value of the industrial area. It was recognized as "National Industrial Heritage" in 2018. Such transformation also plays a positive role in culture. The cultural industry in the eastern suburbs involves music, art, drama, photography, and other art forms, through which culture is widely spread and developed.

### 4.3.4. The Social Effects

The social effects refer to the psychological aspects considering social cohesion, sense of place, community pride, subjective satisfaction, and perception of the multiple benefits of urban regeneration [43,63,64]. Open spaces with greenery, leisure, and entertainment facilities are effective contributors to physical and mental health and social interaction [65]. From the public's cognition and feelings about Eastern Suburbs (Figure 2 and Table 2), we claim that the shared "sense of place" emerges from good demonstration and marketing strategy, not necessarily through bottom-up growth. Although residents and visitors were not fully involved in the reconstruction decisions in the early stage, they were satisfied with the results.

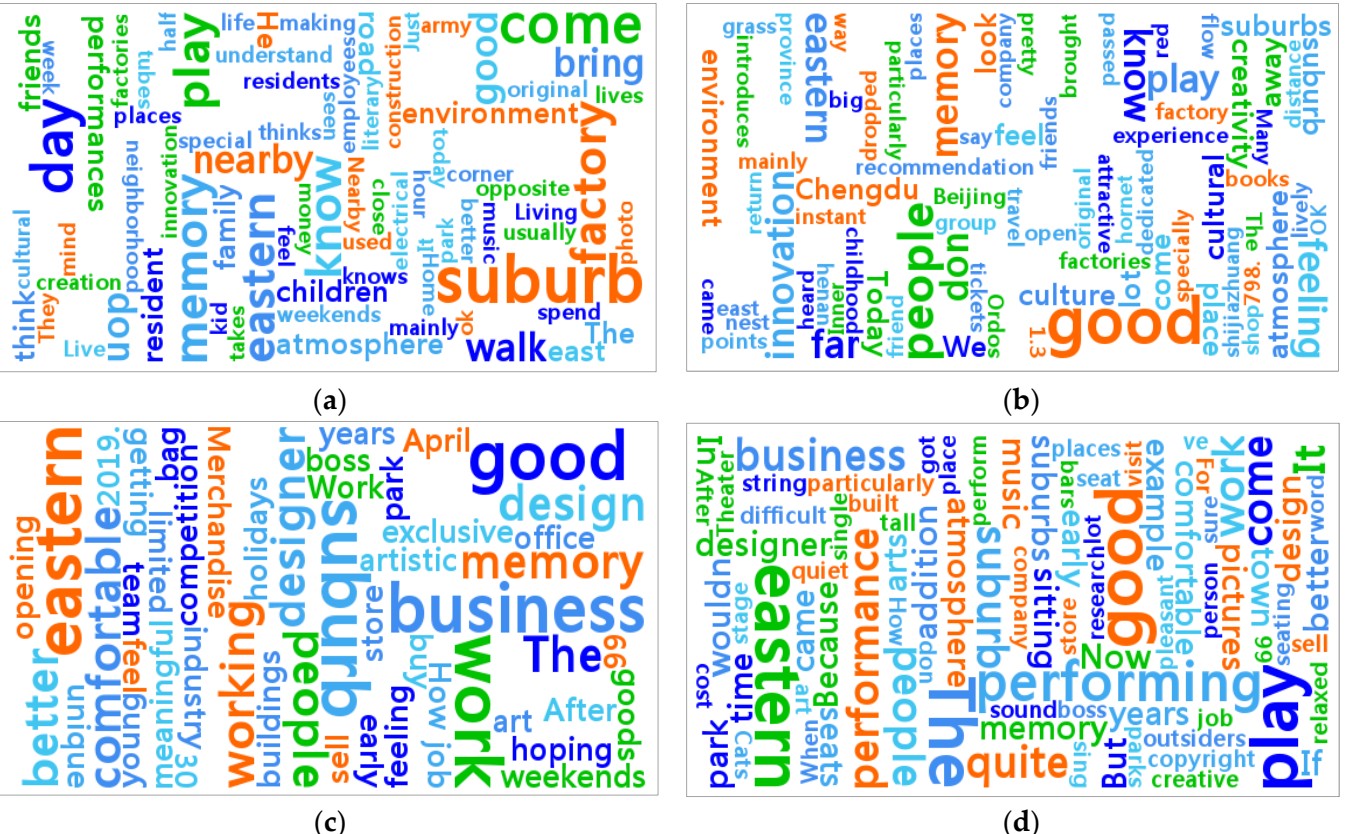

**Figure 2.** Word cloud of stakeholders' subjective feelings: (**a**) the tourists; (**b**) the residents; (**c**) the staffs; and (**d**) the theatre audiences.

**Table 2.** Public's perception of the renewed Eastern Suburbs.

| Interviewees | Subjective Feelings and Descriptions about Eastern Suburbs |
| --- | --- |
| Tourist A | I heard many people talk about this place. I am interested in how the old factories have been transformed. |
| Tourist B | Whether watching shows or taking photos, the environment is good and fascinating. |
| Tourist C | This place is covered with a cultural atmosphere; the transformation is quite good. |
| Tourist D | It's a mecca for taking pictures, with more characteristics. |
| Tourist E | It's very retro here. It is multifunctional—we can come here to watch the show, rather than only hanging out in the park. |
| Tourist F | There is nostalgia and some fresh blood, making the city glow with new glory. |
| Resident A | We grew up here, so we have unique feelings for this place. Coming here is both nostalgic and touching and a good way of entertainment. |
| Resident B | Living nearby, come every week, pleasant environment, nice atmosphere. |
| Resident C | Eastern Suburbs keep the original appearance of the industrial era so that everyone can have a memorial, and the current machine model is the same as before. |
| Staff A | Compared with office buildings, the working atmosphere is relaxing and comfortable. |
| Staff B | Working here is more enjoyable than working in other places. |
| Staff C | Working here is more comfortable and meaningful than working in other places, and always have chance to appreciate the combination of art and goods. |
| Staff D | One of the reasons the shop owner chooses this place is the atmosphere of literature and art. He hopes to sell something artistic. |

## 5. Stakeholder Management in Urban Regeneration

Urban regeneration projects often have a long duration, large scale, multiple steps such as project planning, land acquisition, house demolition, and owner resettlement [3], involving a wide range of stakeholders from various social realms. For instance, the landowners—mainly state-owned enterprises—take the opportunity of the redevelopment of industrial land to obtain higher economic returns; real estate developers are keen to realize the commercial potential; government and urban planners have many social and economic reasons to reuse the underutilized industrial land [19]. The influence of stakeholders on the results of urban regeneration projects can be significant [66]. Due to considerable differences in interests, conflicts between stakeholders occasionally occur [26]. Therefore, the management of stakeholders in urban regeneration is of utmost importance.

### 5.1. The Overview of Stakeholders

The initiation of the Eastern Transfer in 2002 marked the start of the urban regeneration process in a "top-down" manner, with intensive interaction among the city and district government, banks, investors, industrial enterprises and their employees. The critical topic of negotiating was "who should pay" and "how to pay" for the relocation of these enterprises. After several rounds of discussion, a consensus on "land for capital" was reached as a solution for relocation funds. In detail, enterprises obtained funds through land transfer, which could be used for relocation, technological transformation, placement of laid-off workers and retirees, etc. The banks gave priority loans to support enterprises that met the conditions for transformation. After the relocation, the former industrial zone was handed over to the city government, then empowering the developer for reconstruction and operation. Although it was government-led, cooperation and communication among various stakeholder groups were inevitable.

### 5.2. The Government

Most of the urban regeneration projects in China are government-led, in which government plays a leading role in decision-making, such as selection of the relocation site, auction of the original site, renewal planning, transformation investment, and policy support. The locations of the enterprises were transferred to land developers with negotiated prices or the trading price on the land trade market. The government charged 5% of the purchase price or the reappraisal price from the enterprises that transferred before 2003. From this, the government has collected more than 1.2 billion yuan, used for relocation, reconstruction, technical renovation, re-employment, and placement of laid-off workers and retirees. The implementation of this vast systematic project did not increase the financial burden of the city government.

### 5.3. The Relocated Enterprises

Until 2006, more than 90% of the previous Eastern Suburbs enterprises have been relocated to Chengdu Economic and Technological Development Zone, High-tech West Zone, Xindu District, and Qingbaijiang District, as the government suggested and planned. Such transfer has provided new opportunities for enterprises and promoted the economic development of the destination areas. For instance, in Qingbaijiang District, the transferred enterprises have achieved an output value of 9.4 billion yuan in 2006, accounting for 51% of the district's total industrial output value, and contributed nearly 400 million yuan in taxes. The total tax of all relocated enterprises has exceeded 4.5 billion yuan, constituting a long-term stable economic source for the government of the destination areas. The migration of industrial enterprises also produced a multiplier effect—one industrial worker can drive about five to six related service job positions. Through regeneration, existing traditional industries have been consolidated, and new initiatives have been cultivated, promoting the integration of urban and suburban areas.

### 5.4. The Operators and Other Business Participants

There are several business participants involved. Taking Eastern Suburbs Memory Park as an example, Chengdu Media Group is the overall developer and operator designated by the government, responsible for planning, development, management, and operation. It has been using its advantages in media to form the cultural and creative industrial cluster, attracting tourists through a complex combination of cultural heritage visits and various experiential activities. As a cultural resource, industrial heritage has been developed into a tourism industry and has become one of the new industrial activities for economic development in the post-transition region [67]. Cultural and creative industries are interactively coupled and developed [68] and seen as growth drivers for the economy around the world [69]. Under such a circumstance, Chengdu Media Group has driven consumption with numerous caterings, artistic and creative boutiques, commercial music industries, and others, creating business opportunities for many economic participants. Due to the uniqueness and attractiveness of cultural and innovative products, the business environment is ideal for economic participants with a slight pressure of competition.

### 5.5. The Public

We consider the public the workers of relocated enterprises, residents nearby, tourists, and other groups and individuals affected by the regeneration, but have little decision power to influence the regeneration. Laid-off workers of relocated enterprises had opportunities to be re-employed through the development of the tertiary industry, and the government provided policy support for that. Retirees were paid by the relocation enterprises or handed over to local communities for socialized services. The satisfaction of residents, tourists, and other people nearby is also critical to ensure the sustainability of reclaimed urban areas. Residents generally consider the regeneration improved their living environment and created more choices for leisure, family activities, and gathering with friends. They are satisfied with the transformation and have a high stickiness with this area.

For tourists, Eastern Suburb is an attraction site and a multifunctional space providing systemic entertainment items. The feelings and attitudes of the public are essential in the operation phase since they are the main force of consumption. Their interests would affect the business types and modes directly or indirectly.

## 6. Discussion

### 6.1. For What? The Overall Consideration of Urban Regeneration Decision

The decision of Eastern Suburbs regeneration was affected by many factors such as environment, economy, and culture. Before regeneration, the Eastern Suburbs industrial zone and its surroundings were seriously polluted by emissions and sewage. The enterprises were under pressure of economic transformation, facing a profits drop and bankruptcy crisis. In addition, since 2005, the cultural and creative industry was seen as a new path for economic booming in Chengdu, and the protection of cultural sites was vigorously initiated. Therefore, the purpose of Eastern Suburbs regeneration was to improve the urban environment, economy, culture, and other aspects. In such a context, the government seems to be the most effective decision maker with power and resources for such strategic decisions. While allocating resources with its power, the government has made detailed policy arrangements to stimulate socio-economic development with increased stability and decreased uncertainty. Through a series of policies, the government had defined the new sites to relocate enterprises and provided them with favorable policy conditions and supporting resources, such as adjusting or creating new bus routes, providing discount interest on loans, and pollution control subsidies. Fiscal and tax preferential policies also have been issued to promote the commercial development for the new business participants residing in the eastern suburbs after regeneration. Under the government's guidance, enterprises relocated with increased gross value, new operators and business participants achieved good economic benefits, the surrounding environment has been improved, the industrial cultural sites have been protected, and residents have benefited from the improved environment and social services. Various stakeholders have reached a balance of interests regarding the economy, society, and environment. Compared to the bottom-up approach, the government-led top-down approach is more results-oriented and seeks to leverage local knowledge and resources, often in partnership with community leaders and higher-level institutions, and then intervene at the local level [42]. Such an approach could handle complex objectives, organize relative stakeholders, and ensure the effective implementation of the decision. As a result, such strategic decisions strengthen the complementarity of regional advantages, promote coordinated development, and bring new vitality to the city [70].

### 6.2. For/By Whom? The Stakeholders in Government-Led Urban Regeneration

The process of urban regeneration involves different types of stakeholders, each of whom has different roles and distinct needs [71]. The interests of stakeholders could be roughly divided into "administrative and political achievement" (e.g., urban development and social stability), "marketing performance" (e.g., economic benefits), and "community benefit" (e.g., environment, living comfort, fairness, and justice), and often include a combination of two or three of the above [27,72,73]. Eastern Suburbs regeneration is the result of cooperation and communication among various parties. The government has an essential function in the planning and promoting the regeneration process. The government retains strong governance power through its dominant control over policy issuance and resource allocation [74]. Like government entrepreneurship in Western countries, local governments in China dominate urban regeneration processes, control land and taxation, integrate resources, provide public services, and promote cooperation among various organizations, thereby maximizing economic benefits. Drawing on the experience of Western countries, Chinese governments have also used authority and subsidies to attract investment for urban regeneration projects. The enterprises also have the strong bargaining power to realize the plan "land for capital." Large-scale relocation and

reconstruction became possible with the driving force of multiple government policies and positive responses from enterprises. As a result, the relocation has brought development opportunities to the relocated areas and promoted the integration of urban and rural areas. The former employees were properly resettled and actively cooperated with the reconstruction measures. The business partners joined later and played an essential role in operating and maintaining the regenerated sites.

However, failure is also possible in government-led urban regeneration due to the ignorance of institutional foundations. According to Evans, the foundations of urban governance models require the transformation of institutionalized "meta-ideas" (such as administrative norms, legal rules, and other governance mechanisms) that operate at the whole level of society [75]. Sustainable urban regeneration depends on implementing a development model that integrates economic, social, and cultural considerations and resonates with current policy and institutional systems [76]. The role of system and law in urban governance cannot be underestimated. In the case of urban renewal in Turkey, even though the macroeconomic conditions were good and the government was determined to the project, the project still failed because of the unreasonable design of the legal/institutional foundation and the conflicting relationship between different stakeholders. The author argued that the decisions must be made by the local government, which is more aware of the economic, social, and cultural dynamics within its jurisdiction [77]. In the case of urban renewal in Dubai, where the top-down planning model failed to capture the needs of low-income communities, the researchers argued that the state and developers should provide technical and financial assistance to urban residents facing eviction. Another failed case is the Qintailu project in Chengdu, China, where the reconstruction plan was only positioned by the government, regardless of market demand, and the gap between stakeholders and decision-makers led to the failure of the project [57]. Fortunately, the above factors leading to project failure do not appear in our case. The local government has enough administrative power and effectively coordinates the interests of all stakeholders. We argue that top-down is more like a process of governance and decision-making, and does not mean that decision-makers exclude the interests of other stakeholders. Full or partial participation of stakeholders can be integrated and improve the efficiency of urban regeneration. The involvement of various stakeholders should be balanced according to the actual situation. In government-led regeneration projects, policies must be designed reasonably and properly considering stakeholders' interests so that local governments can exercise power according to local conditions.

In addition, Yu et al. found that the policy objectives and priorities of different levels of government in China sometimes differ, such as economic development versus social stability, which in some cases conflict with each other [36]. In the case of Guangzhou Xinyi Club, Li et al. found that the support of the local government and the first-mover advantage of the project supporters are the reasons for the successful development of the project, and the reorganization of industrial land requires pragmatic cooperation between the local government and occupiers and market participants, as well as overcoming decision-making constrained by regulators [19]. In our case, the success of enterprise relocation is closely linked to the proper cooperation between governments at different levels in different regions, and the abundant communication and collaboration with operators and business participants. The completion of urban reconstruction requires coordination at all levels of government to unify goals. It is also necessary for the government and operators to communicate and run in to reach the interests of all parties.

Another thing that does not necessarily cause regeneration projects to fail, but is likely to slow them down, is conflict. It is worth noting that disputes and conflicts often occur, as reported in previous studies. The differentiation of urban and rural land systems in China blurs property rights and increases the number and types of stakeholders, making consensus more difficult [78]. It takes a long negotiation process to determine how to distribute the benefits of urban reconstruction and reach a consensus among core stakeholders [79]. Relocation and compensation arrangements often cause intense conflicts [79,80], sometimes

leading to violent demolition [81–83]. Zhuang et al. addressed this issue by arguing that governments need to consider urban development and human needs, redefine public interests and establish effective dialogue mechanisms [27]. In our case, fierce conflicts did not seem to occur, which may be because the government made decisions beneficial to all stakeholders involved. Government renovation may benefit the public in general, but for affected residents, the focus is more on compensation and relocation [27]. For instance, the former employees of the enterprise were relocated in the form of compensation and re-employment under governments' guidance. Therefore, it is reasonable to consider that making stakeholders foresee the potential benefits effectively alleviates contradictions and conflicts. It can be said that the cooperation among stakeholders determines the success of urban regeneration; nevertheless, the government played the leading role. Due to the immaturity of Chinese civil society, local governments and political leaders have room for discretion through an exclusive decision-making process when implementing policies set by the central government [84–86]. From the perspective of the public itself, people are not always willing to actively participate in urban management decision-making. For example, in the study on community participation in Shanghai, scholars found that some residents expect the government or other parties to arrange everything for them and lack the motivation to participate [87]. The case study of urban heritage in Lijiang also showed that residents sometimes lack strong confidence and participation in local practices [52]. Similarly, the case in Nanshan Industrial Zone showed that most owners have no intention or a wait-and-see attitude [88]. Therefore, we argue that top-down management also needs to take steps to understand the full range of public needs and desires, although only when citizens are partially involved in decision-making. Even in the government-led process, full attention must be paid to the functions and needs of stakeholders to ensure the successful completion of urban regeneration.

### 6.3. How to Proceed? Thoughts on the Best Practice of Urban Regeneration

In China, due to the differences in the geographical, demographic, and socio-economic conditions of urban development, there is no unified, universal, or prescribed form of urban regeneration cooperation [40]. The bottom-up approach has advantages in transparency and equity; however, it has been criticized for difficulties in expansion and replication [89–92], and it is time-consuming and potentially inefficient in dealing with conflicting ideas among multiple stakeholders. We believe that seeking a solution from the perspective of the public and relocating enterprises is inefficient and slow. The relocation of industrial enterprises needs to consider a wider range of issues, such as environmental problems, large equipment depreciation purchase, site, factory staff allocation, business transactions, etc. The abilities and resources of enterprises to find solutions on their own were limited, especially when they were in debt. On the contrary, in top-down decision-making, the government uses its power to keep the planning process under control, rationally allocate resources, and ensure the project implementation in a safe, efficient, and smooth way [74]. The government sees the problem in a much broader context. It can use more funds and conduct better negotiations between different cities, to allocate resources rationally. In our case, the Chengdu government issued "Interim Measures for the Relocation and Reconstruction of Industrial Enterprises in Eastern Suburbs", including measures that banks, financial institutions, municipal public departments, and other relevant institutions should take to promote their cooperation. Subsequent results showed that the decision has indeed achieved beneficial results. Declining enterprises, by relocating to other places, eventually become profitable and drive economic development. Obviously, it is difficult to achieve this in a short time only by the efforts of the enterprise itself. In such cases, a top-down approach may be the most appropriate approach.

Many attempts to include community stakeholders often prove problematic [93–95]. Different stakeholders have different interests and needs, and it is difficult to reach consensus [94]. When the interests of each stakeholder are fully considered, the ability of the whole system to achieve its stated goal of profit maximization is inevitably reduced [95]. Coordi-

nating stakeholders inevitably takes more time and energy, and it is a question whether these time and energy can be paid back and whether they are worth it. These issues need to be considered in a specific context. We further argue that integrating stakeholders' interests are not necessarily linked to participation. When the needs of stakeholders are visible, the government can easily include them in the top-down decision-making sufficiently. Such a point is supported by Sonn et al., who argued that top-down management proved effective at different scales when certain conditions were met [96]. In our case, the public involved may be the factory staff and surrounding residents. Given that the factory is suffering from severe losses and environmental pollution, relocating the factory was a reasonable choice for both staff and residents. Under such circumstances, engaging the public more broadly and profoundly does not necessarily bring out the advantages that previous studies have generally suggested but rather slows down the decision-making process. Although it is criticized that in China, government-led processes are often positioned and characterized by exclusivity, controversy, and unorthodoxy [97], the government-led process can also produce outstanding results in practice if the interests of residents are effectively discussed and fully included [86].

We should be aware that most urban regeneration practices are not at the extreme polar of fully democracy or dictatorship, but on a broad spectrum between the two polars. The urban renewal case of Limmontepe, Turkey, showed that the centralization of planning power in law does not necessarily eliminate the community's ability to participate at the grassroots level [98]. A case study of Lecce in southern Italy, integrating top-down and bottom-up approaches to overcome institutional-level conflicts in the use of urban space. Bottom-up engagement is used to analyze people's vision, which is then shared with decision-makers to develop planning and design solutions from the top down. The case study of Gdańsk, Poland, showed that top-down projects could effectively promote physical regeneration, while bottom-up processes are crucial for social and population regeneration. Therefore, the renewal of urban space requires a mixed strategy [99]. Taking Shenjing Village, Guangzhou, China as an example, the study found that collaborative workshops, as a combination of top-down and bottom-up methods, facilitated joint action between the government and the public through consultation, effectively solving a series of social problems brought about by the rapid urbanization construction from top to bottom [56].

Based on previous studies and our experience, we argue that when implementing top-down urban regeneration, a participatory and inclusive approach could be integrated to fully consider the needs and interests of different stakeholder groups and increase the "buy-in" of the implementation plan [57]. There are various possible ways for stakeholders to participate in a top-down approach: actively or passively, the whole process or only some stages, fully or partially engaged, according to the demand on site (Table 3). The distinction between bottom-up and top-down approaches needs to be updated. Both methods must be considered two mostly coincident moments [100], requiring careful ex-ante planning and ex-post implementation [101], taking the geographical scale, regional structural precondition, sector-specific orientation, and life cycle stage of the cluster into consideration [102].

**Table 3.** Various ways of stakeholder participation in government-led urban regeneration.

| Stakeholders | Manner | | Stage | | Degree | |
|---|---|---|---|---|---|---|
| | Active | Passive | Whole | Periodical | Full | Partial |
| Government | √ | | √ | | √ | |
| Enterprises | √ | | √ | √ | | √ |
| Operator | √ | √ | | √ | | √ |
| Small business | | √ | | √ | | √ |
| The public | | √ | | √ | | √ |

## 7. Conclusions

The urban regeneration approach connects economic growth and environmental enhancement with social and cultural vitality at the regional and local levels. Eastern Suburbs regeneration is a typical government-led process of planning, decision-making, implementation, maintaining, and updating during the last 20 years. By unfolding the process of Eastern Suburbs regeneration and paying specific attention to its stakeholder involvement, we claim that the government-led top-down approach is efficient in large-scale urban regeneration if stakeholders' needs are appropriately considered. The possible ways for stakeholders participation could be selected according to the demand on site. To thoroughly evaluate the regeneration results, it is vital to triangulate the subject assessment done by internal and external stakeholders and the objective assessment using economic, social, and environmental dimensions. There might not be a rule of thumb for stakeholder management due to the dynamics, which need to be carefully considered and discussed case by case. It is also vital to include broader policy and regional development contexts into the negotiation and game system. It should be noted due to the limitation of the case study, careful evaluation should be made before generalizing our findings to other contexts. In the future, more open and equitable stakeholder dialogue should be created from the beginning of urban regeneration with a systematic and comprehensive plan and with support from specific institutional designs, laws, and regulations.

**Author Contributions:** Conceptualization, J.W. and H.H.; methodology, J.W.; validation, J.W.; formal analysis, Y.Y.; investigation, Y.Y. and F.W.; writing—original draft preparation, Y.Y. and F.W.; writing—review and editing, J.W.; visualization, Y.Y. and J.W.; supervision, J.W. and H.H.; funding acquisition, J.W. All authors have read and agreed to the published version of the manuscript.

**Funding:** This research was funded by the National Natural Science Foundation of China, grant number 42101216; Humanities and Social Science Foundation of Ministry of Education, grant number 20YJCZH153, and the Philosophy and Social Science Research Foundation of Chengdu University of Technology, grant number YJ2021-ZD010.

**Institutional Review Board Statement:** Not applicable.

**Informed Consent Statement:** Not applicable.

**Conflicts of Interest:** The authors declare no conflict of interest.

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
