# Peer review of "Stakeholder Management in Government-Led Urban Regeneration: A Case Study of the Eastern Suburbs in Chengdu, China"

_sustainability, doi:10.3390/su14074357_

Round 1

Reviewer 1 Report

The article discusses the benefits and pros of the top-down approach to urban regeneration. To do this, it uses a case study (eastern suburbs of the city of Chengdu, in China). Several questions arise in relation to it:

- In chapter 1, the search for "good practices with global perspectives" is proposed as the objective of the research in reference to the observation of cases. This could be debatable given the peculiar political context in China. There is no similar political-administrative structure in many countries on the planet, nor is there a political regime with so many contradictions in relation to labor, social, economic or environmental issues. Hence, the study of a case in the country may not have the significance formulated by the authors of the document;

- In the methodology section, the qualitative basis seems adequate, as well as the number and type of interviewees. Perhaps certain doubts could arise when interpreting the results obtained in these interviews, especially in relation to the freedom of expression of the government staff and residents. It would be convenient to clarify this question in the article;

- The content of lines 126, 127 and 128 should be deleted;

- Section 4 refers to the actors involved. The problem of the article has to do with the focus of urban regeneration. However, the key issue is not so much the consideration of the benefits of one or another approach but, as indicated in the text, the little decision-making capacity to influence the regeneration of what is called "public". In European or other Western countries, there is freedom of choice of approach. In China, no. That's the real problem. It is not a question of decantation towards one model or another of management, but rather the lack of capacity to choose between one or the other that exists in the country. That would be a key element that should be clearly stated in the article;

- Section 5 is not a discussion. There is no bibliographic reference to support or refute what is stated in the document. Basically, the section is presented as an argument in favor of the analyzed management model. Reference should be made to other international actions based on said model or to other bibliographical references arguing for and against said model. It is raised in a very light way in the introductory section, and it should be developed with more intensity in this section to be considered a true academic discussion;

- In the conclusions, the authors openly state that the government-led, top-down approach is efficient in large-scale urban regeneration if stakeholder needs are properly considered. The key question, in line with what was argued above, is whether it is possible or feasible to launch another model in the country. And if it is, it would be necessary to refer to other urban regeneration projects and interventions in which it has been put into practice. Otherwise, the article becomes a plea for the benefits of a model that cannot be another;

The article is well thought out and well written. The methodological base seems adequate, although it has too many academic inconsistencies. Formulated in this way, it has an excessively descriptive character, not comparative with other management models in the country or abroad. That makes you lose much of the significance of the content of the article.

Author Response

We thank the reviewer for the thorough review. We have revised the paper, and changes are marked in the revised manuscript.

Point 1: In chapter 1, the search for "good practices with global perspectives" is proposed as the objective of the research in reference to the observation of cases. This could be debatable given the peculiar political context in China. There is no similar political-administrative structure in many countries on the planet, nor is there a political regime with so many contradictions in relation to labor, social, economic or environmental issues. Hence, the study of a case in the country may not have the significance formulated by the authors of the document.

Response to Point 1: Thank you for pointing out this issue. We realized that our expression may have caused ambiguity. Our original intention is to provide more cases for the practice of urban regeneration and to replenish the understanding of existing urban regeneration models. The wording may be misleading, and it has been revised with more rigorous wording on Page 2, Lines 69.

Point 2: In the methodology section, the qualitative basis seems adequate, as well as the number and type of interviewees. Perhaps certain doubts could arise when interpreting the results obtained in these interviews, especially in relation to the freedom of expression of the government staff and residents. It would be convenient to clarify this question in the article.

Response to Point 2: Revised. We have clarified the freedom of speech of government workers and residents.

Point 3: The content of lines 126, 127 and 128 should be deleted.

Response to Point 3: It has been deleted.

Point 4: Section 4 refers to the actors involved. The problem of the article has to do with the focus of urban regeneration. However, the key issue is not so much the consideration of the benefits of one or another approach but, as indicated in the text, the little decision-making capacity to influence the regeneration of what is called "public". In European or other Western countries, there is freedom of choice of approach. In China, no. That's the real problem. It is not a question of decantation towards one model or another of management, but rather the lack of capacity to choose between one or the other that exists in the country. That would be a key element that should be clearly stated in the article.

Response to Point 4: Thank you for raising this critical point, and we agree that the choices are limited under the Chinese context. Our case tried to provide some insights on how to bridge the differences between the two paths. We extended the clarification of this issue in both the Literature Review and Discussion. Please find the revision in Lines 147-168, Lines 506-517, and Lines 543-573.

Point 5: Section 5 is not a discussion. There is no bibliographic reference to support or refute what is stated in the document. Basically, the section is presented as an argument in favor of the analyzed management model. Reference should be made to other international actions based on said model or to other bibliographical references arguing for and against said model. It is raised in a very light way in the introductory section, and it should be developed with more intensity in this section to be considered a true academic discussion.

Response to Point 5: Thank you for your valuable suggestions, and we have extended and improved our discussion. Relevant literature has been cited, and cases from other countries have been discussed in Section 6.

Point 6: In the conclusions, the authors openly state that the government-led, top-down approach is efficient in large-scale urban regeneration if stakeholder needs are properly considered. The key question, in line with what was argued above, is whether it is possible or feasible to launch another model in the country. And if it is, it would be necessary to refer to other urban regeneration projects and interventions in which it has been put into practice. Otherwise, the article becomes a plea for the benefits of a model that cannot be another.

Response to Point 6: Thank you for pointing out this critical issue. We realized that our points were not clearly stated in the original manuscript. In the revised version, the advantages and limitations of different approaches are extensively discussed concerning our case in the Discussion section. To keep the conclusion concise, we have raised our argument more clearly in the Discussion section.

Reviewer 2 Report

While Chengdu is an interesting case for studying urban regeneration, this paper did not add much to the literature. The argument is loosely organized and the major research question is not clearly stated. Whether do the authors focus on the process or the outcome of regeneration? What are the new findings compared with previous studies?

Another problem is this paper did not provide much information supported by the interviews, making it not sufficiently material-supported.

Author Response

We thank the reviewer for the thorough review. We have revised the paper, and changes are marked in the revised manuscript.

Point 1: While Chengdu is an interesting case for studying urban regeneration, this paper did not add much to the literature.

Response to Point 1: Thank you for pointing out this issue. In order to contribute to the current literature, in the revised version of the manuscript, we have extended our discussion section. We proposed in the discussion section that the choice of top-down and bottom-up approaches should be made considering specific circumstances, including political background, government power, willingness to participate in public participation, and the ability of the public to make independent decisions.

Point 2: The argument is loosely organized and the major research question is not clearly stated.

Response to Point 2: Thank you for pointing out this issue. The main research question has been stated in Line 99-101.

Point 3: Whether do the authors focus on the process or the outcome of regeneration?

Response to Point 3: We pay more attention to the stakeholder participation process to bring a better outcome. We have extended the discussion to express our concern on the process.

Point 4: What are the new findings compared with previous studies?

Response to Point 4: Revised. We extended the discussion section with more detailed information on our case compared to various case studies worldwide. We clarified the pros and cons of different approaches and supplemented the method integrating stakeholders’ interests.

Point 5: Another problem is this paper did not provide much information supported by the interviews, making it not sufficiently material-supported.

Response to Point 5: Thank you for pointing out this issue. The entire Section 4 is formed using the information gathered from our interviews. However, in order to keep the writing smoothly flowing, we did not state what the interviewees said as direct quotations.

Reviewer 3 Report

The theme is interesting but needs some improvements:
The reader should know better the macroeconomic environment of the sample between 2000 and 2020. Was there any reason for choosing the time period? Has the economy not changed in 20 years?
Literature on the subject is scarce and should be reinforced in its own section.
What was the gap in the literature that the authors identified for carrying out this research?
The cultural and social aspects have to be better developed as well as the effects on the economy. The authors were sparse in this explanation.
Authors should reinforce the explanation on the adequacy of the methodology applied to data treatment.
The discussion of results should be improved and criticized in terms of other authors.
the authors have to admit the limitations, both of the sample and of other types of variables to be considered in the interviews. The theoretical implications, as well as the practical contributions of this work, are lacking.

Author Response

We thank the reviewer for the thorough review. We have revised the paper, and changes are marked in the revised manuscript.

Point 1: The reader should know better the macroeconomic environment of the sample between 2000 and 2020. Was there any reason for choosing the time period? Has the economy not changed in 20 years?

Response to Point 1: Revised. We explained in section 4.1 that the macro background of the past two decades is that the industrial economy has begun to sink, while the cultural and creative industry has become a new economic growth point. And in the micro-level, the regeneration of Eastern Suburbs initiated around 2000, and its effects gradually show in the following years. Our field study was conducted from January to April 2020. Therefore, we choose the period 2000-2020 to reflect the process and effects of the case study.

Point 2: Literature on the subject is scarce and should be reinforced in its own section.

Response to Point 2: Thank you for your suggestion. We have rewritten the theoretical part with a dedicated Literature Review section.

Point 3: What was the gap in the literature that the authors identified for carrying out this research?

Response to Point 3: Thank you for your question. We have revised and identified the gap in the Literature review and stated our research question in Line 99-101. The advantages and disadvantages of top-down and bottom-up approaches are widely discussed in the literature, but the best application scenarios are rarely mentioned. And the experiences from western countries might not be applicable in China. Such gaps enlightened us to explore the stakeholders in the process of government-led urban regeneration in the Eastern context. Therefore, we raise our research questions: How did stakeholders participate and collaborate in the reconstruction process? Is top-down government leadership effective? How to integrate stakeholders’ interest in the government-led process?

Point 4: The cultural and social aspects have to be better developed as well as the effects on the economy. The authors were sparse in this explanation.

Response to Point 4: Revised. We have further explained the cultural, social, and economic impact in Section 4.3 and in Section 6 Discussion.

Point 5: Authors should reinforce the explanation on the adequacy of the methodology applied to data treatment.

Response to Point 5: Revised. We have explained data processing in more detail in Line 193-197.

Point 6: The discussion of results should be improved and criticized in terms of other authors.

Response to Point 6: Revised. We have extended our discussion, including comments on other researchers' views, the links between different opinions, and reasons for/against both top-down and bottom-up models.

Point 7: The authors have to admit the limitations, both of the sample and of other types of variables to be considered in the interviews.

Response to Point 7: Thank you for your suggestion. We have revised and acknowledged the limitations in Section 3 Methods and Section 7 Conclusion.

Point 8: The theoretical implications, as well as the practical contributions of this work, are lacking.

Response Point 8: Thank you for pointing out this issue. We have described our practical contributions in the Discussion section, including the applicability of top-down approaches in eastern contexts and different ways to manage stakeholder participation, which enriches understanding of urban regeneration in the Eastern context and answers the research questions set at the beginning of his paper.

Reviewer 4 Report

The strengths of the article can undoubtedly be considered the choice of topic and its processing at the level of the Eastern Suburbs in Chengdu, China. The government-led top-down urban regeneration approach the authors surprisingly consider as an opportunity to bring high-quality results for large-scale post-industrial transformation with inclusive stakeholder management with several possible ways to involve stakeholders in the whole process or some stages, and engaging fully or partially, according to the demand on site. The authors of the scientific article approach the study of the issue in an academic way. They are based on relevant theoretical backgrounds, works, and research by foreign authors evaluating and summarizing attitudes to the issue of possibilities, and overall the understanding of the government-led top-down urban regeneration approach. This approach is to some extent in order and is often used in theoretical analyzes of economic issues. In the empirical part, the methods used were well selected for this type of research, a case study using semi-structured interviews of selected actors, appropriately approaching the issue. However, what I see as problematic is that the authors did not set research questions, so it is impossible to state if they were answered.
I note that the conclusions are very general. Given the important issues addressed, I would recommend formulating the conclusions with regard to the previous discussion to formulate more specific or supplement with specific proposals.
Otherwise, in principle, I have no reservations about the article, and after incorporating the stated comments, at least at the level of defining the research questions and their answers in the results section, I recommend publishing the article.

Author Response

We thank the reviewer for the thorough review. We have revised the paper, and changes are marked in the revised manuscript.

Point 1: What I see as problematic is that the authors did not set research questions, so it is impossible to state if they were answered.

Response to Point 1: Thank you for pointing out this issue. We present the research question on Lines 99-101 as: How did stakeholders participate and collaborate in the reconstruction process? Is top-down government leadership effective? How to integrate stakeholders’ interest in the government-led process? We addressed the first and second question in Section 4 and 5, and addressed the third questions in Section 5 and 6.

Point 2: I note that the conclusions are very general. Given the important issues addressed, I would recommend formulating the conclusions with regard to the previous discussion to formulate more specific or supplement with specific proposals.

Response to Point 2: Revised. We have refined the conclusions based on the content of the Discussion section.

Round 2

Reviewer 1 Report

The adaptations to the original document have been numerous and have responded to what was stated in the previous review.

Author Response

Thank you for your comments which help us a lot to improve the quality of the manuscript! We appreciate that very much!

Reviewer 2 Report

While the authors did comprehensive revision, they seem only to add some literature rather than original information. The major arguments are still not clear.

Author Response

Point 1: While the authors did comprehensive revision, they seem only to add some literature rather than original information. The major arguments are still not clear.

Response to Point 1: Thank you for your comments. We have improved our manuscript by (1) clearly stating the research questions; (2) organizing the research findings according to the research questions; (3) adding new literature to widen our discussion and reflection; (4) adjusting the structure of the argumentation. 

Reviewer 3 Report

The Authors have made an effort to respond to all comments and suggestions, so I believe that the article meets the conditions to be published.

Author Response

(The authors gave the same response as above.)
